# Lean-Interaction: passive image manipulation in concurrent multitasking

Danny Schott*
Otto von Guericke University

Benjamin Hatscher†
Otto von Guericke University

Fabian Joeres‡
Otto von Guericke University

Mareike Gabele§
Otto von Guericke University

Steffi Hußlein¶
Magdeburg-Stendal University of Applied Science

Christian Hansen‖
Otto von Guericke University

## ABSTRACT

Complex bi-manual tasks often benefit from supporting visual information and guidance. Controlling the system that provides this information is a secondary task that forces the user to perform concurrent multitasking, which in turn may affect the main task performance. Interactions based on natural behavior are a promising solution to this challenge. We investigated the performance of these interactions in a hands-free image manipulation task during a primary manual task with an upright stance. Essential tasks were extracted from the example of clinical workflow and turned into an abstract simulation to gain general insights into how different interaction techniques impact the user's performance and workload. The interaction techniques we compared were full-body movements, facial expression, gesture and speech input. We found that leaning as an interaction technique facilitates significantly faster image manipulation at lower subjective workloads than facial expression. Our results pave the way towards efficient, natural, hands-free interaction in a challenging multitasking environment.

**Keywords:** Multimodal Interaction; Multitasking; Hands-free Interaction; Radiology; Medical Domain

**Index Terms:** Human-centered computing—Human computer interaction (HCI)—Interaction techniques—; Human-centered computing—Human computer interaction (HCI)—Interaction paradigms

## 1 INTRODUCTION

Multitasking is an essential part of today's working environments, whether intentionally or involuntarily through a plethora of devices and communication channels. In this context, working on a screen while interruptions appear occasionally is a widely investigated scenario [3, 16, 22].

Even though it is important to understand how recovery from interruptions work, there are scenarios where the primary task is essential and cannot be interrupted completely, but still, additional information is required to reach the overall goal. Prominent examples for such demanding scenarios are flight coordination [17], piloting [15], train driving [13] or medicine [20]. The focus here shifts from recovery from distraction to how to maintain an acceptable level of performance for a specific task during concurrent multitasking.

To find suitable input methods that fit these needs, we propose a 3-step approach: First, an exemplary scenario from the medical context is chosen and investigated in-depth to understand the restrictions

*e-mail: danny.schott@ovgu.de
†e-mail: benjamin.hatscher@ovgu.de
‡e-mail: fabian.joeres@ovgu.de
§e-mail: mareike.gabele@ovgu.de
¶e-mail: steffi.husslein@hs-magdeburg.de
‖e-mail: christian.hansen@ovgu.de

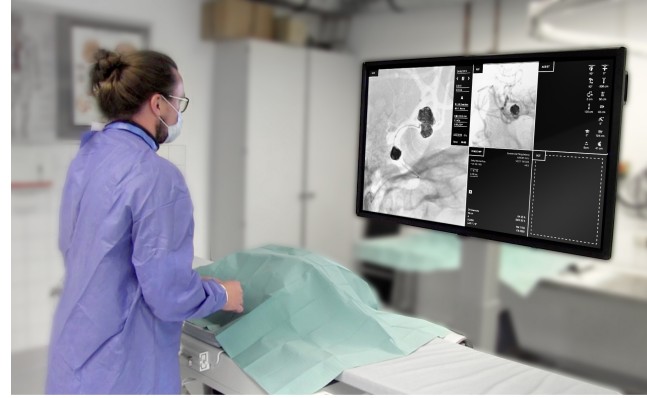

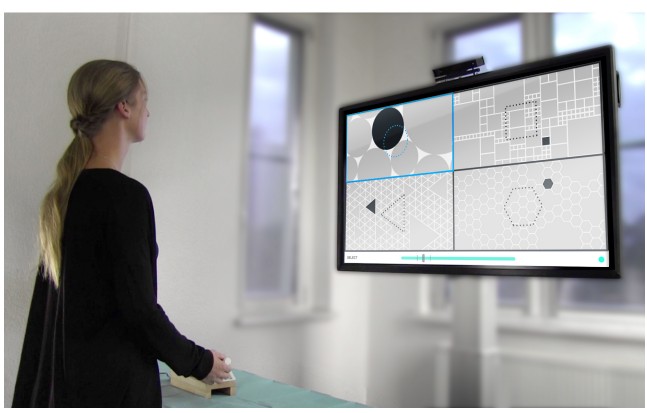

Figure 1: We took the complex activities of a radiologist as a starting point to differentiate between primary and secondary tasks (left picture). We created a simulation of this scenario by abstracting these tasks and developed natural input modalities for hands-free control (right picture).

and requirements of a representative real-world worst case as well as the occurring combinations of tasks. Second, input methods that potentially require notably low cognitive resources are selected, informed by the multiple resource theory [25], the passive input paradigm [21] and observations. Third, the scenario is abstracted to minimize the influence of domain knowledge and evaluated in a user study using a between-subject design.

The main contributions of this work are the development, abstraction, demonstration and evaluation of a free-hand input modality for Zoom and Pan based on natural behavior with minimal influence on a primary task. This may lead to safer, faster and cheaper use in demanding interaction scenarios.

Analytical observation has revealed which natural movements are suitable as complementary input modalities. When overviewing a picture, the participants mainly moved their upper bodies forward and backward, while simultaneously moved their eyebrows up and

down. Within the evaluation of a prototypical setup, it has been shown that leaning is suitable as a secondary input modality since it has minimal influence on the primary task. A secondary input on the movement of the eyebrows, in contrast, proved to be unsuitable as manipulation technique. It's less accurate and physically demanding.

## 2 RELATED WORK

### 2.1 Concurrent Multitasking

Performing two or more tasks in parallel or short succession is called multitasking. Salvucci et al. proposed a continuum ranging from concurrent interaction to sequential multitasking [23]. Adler and Benbunan-Fich found that the degree of multitasking influences productivity and accuracy differently: medium multitaskers are more productive than low or high multitaskers, while accuracy decreases with the degree of multitasking [2]. Interruptions lead to additional time to resume the primary task [18, 24] and are more disruptive when occurring at points of a higher workload than during low workload phases [1]. It comes as no surprise that users tend to defer interrupting tasks to periods of lower workload [22]. A dual-task setting by Janssen et al. concludes that people can strategically focus their attention on certain tasks in order to achieve a certain performance and that choice of strategy has a major impact on task completion [10,11]. Wickens' multiple resource theory assumes that multiple cognitive resources are tied to different modalities and tasks only conflict if they tap into the same resource [25]. This implies that the brain is able to separate different cognitive processes from each other. These findings lead us to the idea that if the influence of a secondary task on a concurrent primary task should be minimized, investigating multimodal interaction approaches seems plausible.

### 2.2 Multimodal Input

According to Oviatt, human-computer input modes can be separated into active and passive ones [21]. Active modes require intentional action by the user, while passive modes rely on unintentional action or behavior. Using passive modes as input lowers the cognitive effort as no explicit command has to be given by the user. This idea connects to non-command interfaces by Nielsen, which derive user intentions from observing the user [19]. With the goal of minimized influence in primary task accuracy in mind, it could be hypothesized that passive input modes reduce the influence of the secondary task even further than active input modes.

## 3 MULTITASKING IN INTERVENTIONAL RADIOLOGY

To investigate the idea of multimodal input for concurrent multitasking scenarios, we picked interventional radiology as a concrete example. During an intervention, interaction with medical images is required simultaneously to the high-priority task of instrument handling. During radiological interventions in general, a needle or catheter is inserted through small incisions into the patient's body. In order to navigate the instrument to the targeted pathological structures to be treated, the radiologist relies on real-time image data gathered by imaging modalities such as magnetic resonance imaging (MRI), computed tomography (CT), ultrasound (US) or accesses images recorded before the intervention [14]. A common workaround in clinical practice is installing an assistant as a proxy user to maintain asepsis (and due to workflow restrictions), which is time-consuming and error-prone compared to direct interaction [5, 9, 20] and is rated by clinicians as least practicable [26].

To gain insights into challenges in clinical practice we observed a radiological intervention and conducted a semi-structured interview with an experienced radiologist regarding the challenges of human-computer interaction tasks. In the following section, we report observations, expert comments and conclude difficult secondary tasks.

During the observation, the radiologist wanted to take a closer look at the details of the latest x-ray image. A range of strategies to achieve this was observed during the intervention.

- **Leaning closer:** The radiologists leans over the patient table towards the display showing the image.

- **Moving the display:** An assistant is asked to move the display closer to the physician.

- **Pointing gesture:** While examining the image, the radiologist points at a specific spot on the screen without touching it and instructs an assistant to enlarge this area of the image.

- **Verbal task delegation:** When both hands are occupied with guiding the catheter, an assistant is instructed verbally to adjust the image section displayed on the screen.

The expert interview aimed at understanding concrete action sequences concerning image selection and zoom was conducted to investigate the potential for direct interaction methods. Overall, the radiologist assessed the operation of the current system very positively, but restrictions in the operation of the control panel were often perceived as disturbing. It is particularly interesting to note that when navigating through the images, there is no need to look at the joystick, which suggests that hand-eye coordination has been perfected by routine. It was pointed out that different layouts of the x-ray images are used according to the user's preference. The radiologist interviewed, for example, prefers a permanent full-screen view in order to be able to see details better on the screen. In this context, he emphasizes the necessity and current problems of zoom functions: "Definitely necessary [...] In principle, zooming is possible, but it is connected with a lot of manual actions and therefore not popular among us". In the angiography suite used for the observed intervention, this function is not available quickly enough. The desired image segment must first be selected, then navigated to the image settings with the help of a touch interface, whereby the view can then be enlarged by 150%. Afterward, the image section can be moved with a joystick. Zooming, in his case, takes place in fluoroscopic 2D as well as in 3D volume data. The respondent himself usually performs this task, because a description of the target to the MTRA would be too inefficient. For an efficient operation of the system, "[...] Infinitely variable magnification and pan is indispensable for a useful zooming function".

## 4 CONCEPT DEVELOPMENT: ABSTRACTING MEDICAL TASKS

### 4.1 Catheter navigation as primary task

The task referred to below as "primary task" simulates the medical task of a physician during a radiological intervention. In this context, the task has the highest priority for the physician, because his primary goal is to perform the intervention and to ensure optimal care for the patient. During a catheter intervention, instruments must be held and guided to the correct position in the body. The correct alignment by moving the tip back and forth requires a lot of concentration and is made more difficult by the patient's breathing movements or movements.

### 4.2 Abstraction

Our implementation aims to create a situation in which the user performs a motor activity with their hands. The manual movements of a radiologist are to be imitated using the example of the guidance of a catheter in a vascular structure during an intervention. We simplified this task to forwards and backwards movements and created a prototypical input device (see Figure 2). An abstract visualization in form of a horizontal bar displays the user input and a target marker. A vertical line represents the current position of

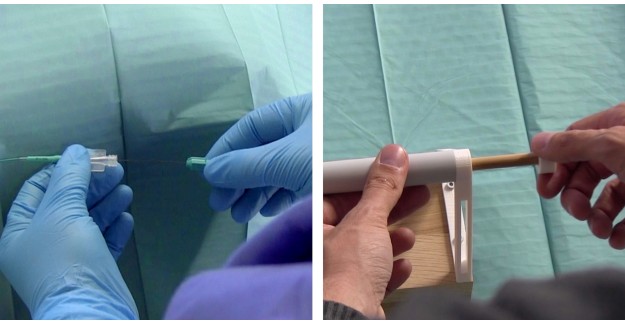

Figure 2: Left: Catheter Navigation Task. Right: Prototype for the simulation of that task.

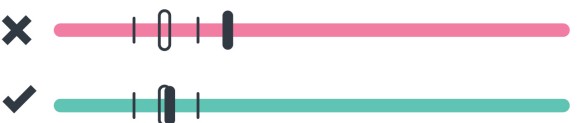

Figure 3: Slide control for transmission and feedback of the user input. The dark line element indicates the direction. If it is within the tolerance range (thin lines / contour), the display is green (bottom). Outside this range, the display turns red (top).

the input device. A marker with a range of tolerance moves in a specific sequential movement along the x-axis and forces the user to pay attention to this task all the time (see Figure 3). To force the focus on the primary task even further, the marker jumps to a random position on the bar every 10 seconds.

### 4.3 Image Manipulation as a secondary task

Current angiography systems offer the possibility to interact intra-operatively with image data on a monitor. Usually, one or more monitors are located directly in the radiologist's field of view, where various information such as live and reference images, patient data and system parameters are displayed. The displays are divided into different areas with an individually configurable layout. The selection of elements, the change of different views and modes, calling certain functions and the manipulation of images, such as scrolling in data sets, changing contrasts or adjusting the magnification factor are common interactions in such systems. In our work, these actions are summarized under the term secondary task. This secondary task can be seen as combinations of two fundamental interactions:

#### 4.3.1 Selection

Within the system, medical image data sets or viewports on the screen can be selected, functions executed or modes changed. The desired object or view on the monitor interface is selected first and the selection is then confirmed. The modalities which are available in current systems are located on the control panel which enables one- or two-dimensional interaction by means of keys, joysticks and touch screens.

#### 4.3.2 Manipulation

The selected data can then be manipulated in one or two dimensions using the same control elements. Frequent tasks are scrolling through image series, magnification of specific structures (zoom), shifting the visible image section (pan) and changing the image contrast and

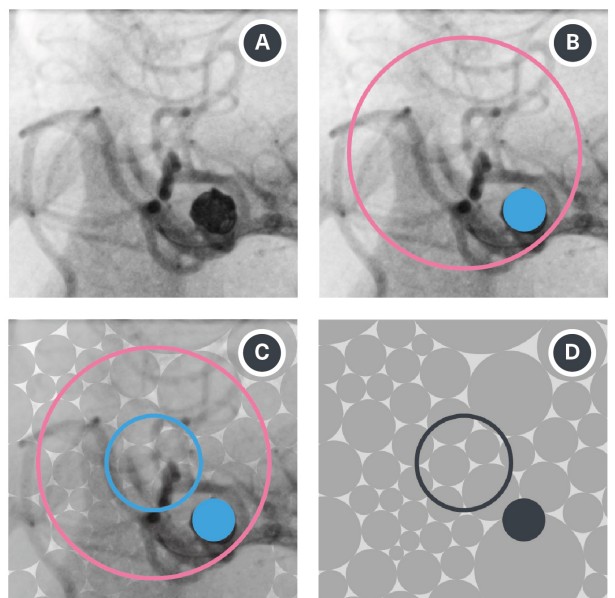

Figure 4: Process of visual abstraction of medical imagery.

brightness (windowing). Further, these input methods are used to rotate and enlarge 3D volume data sets.

### 4.4 Abstraction

During our clinical observation, the interaction steps in the treatment of an arteriovenous malformation (AVM) in the brain turned out to be the foundation for our secondary interaction task. An AVM is a congenital abnormal tangle of blood vessels in which arteries are directly connected to veins, disrupting normal blood flow and oxygen supply. It can be visualized with imaging techniques such as digital subtraction angiography (DSA). The radiologist's goal is to locate these and treat them accordingly. Fluoroscopic X-ray images often have relatively low contrast and low resolution. It is difficult to precisely recognize details such as the position of the catheter within such images and to distinguish overlaying blood vessels. To get as most out of the images as possible, the radiologist should be able to zoom and pan into the region of interest.

We abstracted a typical image recording of an AVM, which can be seen in Figure 4.[A] shows an original recording of an AVM . In the next step [B], the size of the entire structure (red line) and the size of the AVM (blue circle) were marked. In [C] the region of interest is located between the viewfinder (blue line) and the whole structure (red line). Only within this radius were randomized target elements distributed and during the task, the region of interest must be brought into the target position. The surrounding vessels might serve as orientation features when panning and zooming and are replaced pattern of geometric shapes in shades of grey. Four different patterns (ellipse, rectangle, triangle, hexagon) were designed to simulate different images. The abstracted task view can be seen in image [D].

## 5 INTERACTION METHODS

In the following, direct interaction techniques that allow performing the secondary task while the hands are occupied are described. We focused on interaction methods that do not occupy the hands as they are used to perform the medical, primary task, the feet as foot pedals are a standard method to control angiography systems and body-worn equipment such as sensors or head-mounted eye-tracking

devices are ruled out due to sterility concerns. Hatscher et al. found that contactless interactions via speech or gesture are suitable, but that interaction with the feet is most effective and also most accepted by the users [7]. Johnson et al. also describes that foot pedals in the OR are used to trigger image capture and that both these and the hands (holding and manipulating the wire and catheter) are extremely busy, which is why we have decided not to strain these extremities any further [12].

## 5.1 Selection

Selecting a specific element on the graphical user interface can be divided into subtasks pointing and confirmation. Head movements are used to allow pointing without using the hands or fingers. Since only a coarse selection of a dedicated range is necessary, changing the head direction serves as a pointer tool. A cursor's position is directly mapped to the direction the face is pointing. This corresponds to the natural way of turning the head towards an object of interest, like it is used for example for head mounted displays [14]. Confirmation of the element to be pointed at is done via voice commands and head gestures.

The voice command to confirm a selection is "select", which has to be uttered while the head-controlled cursor points at the desired object. To release the current selection, the user has to say "exit". For a completely hands-free workflow, the commands "start" for system activation and "stop" for deactivation act as a clutching mechanism to avoid unintentional input [4].

Confirmation of a selection with head gestures is done with a nod. Similar to voice commands, the cursor has to stay over the object to be selected while performing the gesture. Shaking the head deselects the current object. Both head gestures are further used for activation and deactivation. An overview of all selection functions and corresponding input methods can be found in table 1.

Table 1: hands-free selection techniques

| Selection | Voice commands | Head gestures |
|---|---|---|
| System activation | "Start" | Nod |
| Selection (Point + Confirm) | Head pointing + "Select" | Head pointing + Nod |
| Deselection | "Exit" | Shake |
| System deactivation | "Stop" | Shake |

## 5.2 Continuous Manipulation

For image manipulation, we focused on panning and zooming as it corresponds to 1D and 2D input and therefore, can be transferred to a wide range of HCI tasks. The choice of manipulation techniques lies at the root of the observations in the medical context. In this case, for example, hands-free input can be achieved by leaning and with facial expressions. Other possible inputs, such as from the shoulders, may be less suitable for these purposes, as they can have a greater impact on the task.

Based on our observations during a radiological intervention, leaning closer to a display indicated the need to see more details. Therefore, leaning to the front is used for zooming in while leaning back zooms out. The zoom level is mapped on the leaning angle of the user. The position of the upper body is set as 100% magnification (no zoom) when the desired view is selected by using one of the selection methods described in Table 2.

Lowering and lifting the eyebrows is used for zooming in a similar fashion. By trying to recognize something in the distance, you lower your eyebrows to focus on the target and recognize it more clearly. Lowering starts zooming in while raising the eyebrows zooms out as the position of the eyebrows is similarly hard to control and to measure. Due to the low resolution of this movement, this method applied a fixed rate of change if the eyebrows were lifted over a certain upper threshold or below a lower threshold. During both methods, panning can be performed simultaneously by head pointing.

Both manipulation techniques are based on passive actions which, according to Oviatt [21], are defined as unintentional behaviour. We have taken this as an opportunity to transform these passive actions into active interaction techniques so that the interaction corresponds to the natural behaviour.

Table 2: hands-free manipulation techniques

| Manipulation | Full body movement | Facial expressions |
|---|---|---|
| Zoom in Zoom out | Lean to front Lean to back | Lower eyebrows Raise eyebrows |
| Pan | Head pointing | Head pointing |

## 6 EVALUATION

### 6.1 Study design

We manipulated three independent variables. First, we varied whether participants performed the concurrent primary task (PT) while interacting with the images to investigate the influence of the task on the interaction methods and vice versa. The presence of a primary task was varied between subjects. The second variable was the frame selection method (speech-based or gesture-based). Finally, the image manipulation method varied between leaning manipulation and eyebrow manipulation. The latter two variables were within-subject variables. We investigated three dependent variables. Task completion time was recorded for each trial. We also measured the proportion of time spent outside the primary task target area (hereafter called the error time). We assessed this as an indicator of primary task performance. Finally, we analyzed the overall unweighted NASA-TLX rating (RAW TLX) as an indicator of the subjective perception of the interaction concepts.

We divided the participants into two test groups. Test group A tested all conditions of the experiment with the primary task, while test group B went through the same conditions without the primary task. One input modality for selection (head gestures or voice) was combined with one input modality for manipulation (full body movement or facial expression). Each subject underwent four possible modality combinations to perform the secondary task. The assignment of the combinations and sequence in the execution was randomized so that if possible, no identical modality follows another.

### 6.2 Participants

Our goal was to gain general insights into the developed hands-free interaction techniques and their interaction concerning cognitive and physical stress in a scenario similar to surgery. For this reason, we selected a heterogeneous group of participants with medical, technical and creative backgrounds.

We recruited 16 participants (10 female; 6 male) in the environment of our university-aged between 22 and 38 (M = 26.9; SD = 4.3). The user study last between one and one and a half hours and participants received between 15 and 30 Euro (Due to recruitment problems, the remuneration had to be increased for half of the total participants; mainly medical students).

Seven subjects were students of human medicine - the others had a technical or creative background. Regarding their background, the participants were equally assigned to the test groups with and without primary task. Seven out of eight subjects in group A indicated the right hand as dominant. This information was necessary because the system had to be adjusted accordingly, and the instrument had to be aligned accordingly. It was also ascertained whether there was a

speech disorder in order to check for possible complications with speech input, whereby all test persons denied the question. Further, the participants were asked about a visual disorder, whereby only persons whose defective vision is not too pronounced were invited because a spectacle frame can partially impede the interaction with the eyebrows. Through self-testing, it was possible to say in advance that the system could be operated without restriction in the case of low short-sightedness. Six people reported a visual impairment and one person a color and vision impairment. Further information limited knowledge in the areas of human-computer interaction, gesture control, tracking, and voice control. Here it should be examined whether and if so, what influence the respective skills and knowledge have on the system in general. The survey was carried out using a Likert scale from none (1) to very experienced (5) - based on experience in the respective areas.

## 6.3 Participant task

In the starting point, the participant sees four equally sized segments arranged in a grid, which accommodate a pattern of different geometric forms and a dashed contour of the respective form in its center. A dark, semi-transparent surface lies above the segments and signals a standby mode. In the lower area of the interface, a status light indicates whether a user has been tracked and a text field shows which input has been made. Once a user has been tracked, a cursor is displayed that can be moved by changing the head direction. After entering the initial start command (head gesture or voice), the interface clears up, the input mode displays the current status "Start", and the slider is activated and can be moved. In addition, individual shapes from the background pattern appear dark due to a random selection by the system. It is now possible for the user to select a segment by a selection method (head gesture or voice), on which he points with the cursor. When the command is entered, the selection of the area is highlighted with a blue frame and the word "Select" appears in the area of the input mode. The participant is now directly in zoom mode and can manipulate the image. At the same time, the cursor disappears because it is the same as the geometric contour. The user enlarges the texture in the respective segment by the respective manipulation technique (leaning or eyebrow), intending to bring the respective dark geometric form to the size and position of the viewfinder (dashed contour). At the same time, the user moves the slider, which visualizes the correctness of the input by a color change between green and red. When the position and size match, the contour disappears, and the dark element shows a blue color that the task is completed. The blue frame remains until the deselection command "Exit" is entered, which is also displayed in input mode. After this input, the cursor appears again. As soon as all elements in the four segments have been enlarged and the user is in selection mode, he can terminate the system or task with a final command (head gesture or voice). The input mode shows the word "Stop", the upper area is faded over again, and the slider is inactive. The essential system states are summarized in Figure 5.

## 6.4 Setup

### 6.4.1 Software

The software prototype was implemented in Unity 3D 2018.2 using C#. The voice and gesture control was realized with the included SDK (Software Development Kit) of Microsoft Kinect v2.

### 6.4.2 Body-Movements

The Kinect provides camera coordinates to find 3D points in space. In addition to color and depth information, it also provides integrated skeleton tracking for capturing the human body, which generally determines whether a user is located and in what position he or she is relative to the room. If an input were made for selection (head gesture or voice), a zero point would be set. If the user leans forward from this point to a maximum of 40 cm (about 20°), the image is

enlarged (maximum magnification level 300%). This leaning angle is similar to the one we observed and ist mapped to a rate of change, i.e. straight pose == no zoom; leaning forward a little == zooming in slowly; leaning forward a lot == zooming in fast. Hann et al. present a similar zoom technique in a virtual reality concept for the GI endoscopy [6].

### 6.4.3 Facial Expressions

For the interpretation of the facial expressions, the library used can be used to access aspects that were used to implement the interaction with the eyebrows. Three points on the face were recorded for this purpose: The center of the left and right eyebrows including nose tip, whereby the difference of the distance between the two eyebrows and the nose tip was determined.

### 6.4.4 Head Movement

By rotating the head, the user can move a cursor and also use this method to pan in images. Direction vectors were created from the orientation coordinates to implement the head gestures. Vector changes in an interval were determined and different states checked whether there was a change in the head position to recognize a head gesture (nodding and shaking). Thus the physical strain was kept as low as possible because even minimal movements of the head could be detected. The received data were interpolated by exponential smoothing into smoother movements.

### 6.4.5 Voice Recognition

A list of the defined signal words was created ("Start", "Select", "Exit", "Stop") and we used the Kinect SDK's language package for keyword detection. The confidence level for keyword recognition was set to low throughout, which meant that even speech input by non-native speakers was recognized consistently well.

### 6.4.6 Data Log

A data logger has been integrated, which enables the display of command inputs and outputs. The values are stored in a log file so that they can be converted into tables for later evaluation. At the time when the user starts processing the secondary task by initial execution of the respective interaction, a time counter starts until the user completes the task by a command (task completion time). Furthermore, the incorrect input (time out) of the user was recorded, i.e., the time how long the input was outside the tolerance range.

### 6.4.7 Hardware Prototype

The physical prototype was implemented with the physical computing platform Arduino (model Uno R3), its associated development environment and a distance sensor. The sensor sits in a tube and measures the distance in a range of 0 - 100 mm with a resolution of 1 mm (deviation 3%) to the rod guided by the user. A displayed slider in the interface transferred the user input 1:1 via USB to Unity. The user moves a vertical line in horizontal direction with the aim to keep this line within a defined tolerance range. As long as the line is within the range, the slider bar remains green. If this is left, it turns red. The components were brought together into stable housing using 3D printing.

The experiment was carried out in a laboratory with one subject and one investigator. Similar dimensions were chosen to simulate the overall structure of an operating theatre. The subject was exactly 180 cm away from a 75" monitor, measured from the edge of the table to the display. Above the display, a Microsoft Kinect was installed, which was adjustable in angle. In between was a table on which the hardware prototype for the primary task was located. The monitor and table were height-adjustable and adapted individually to each test subject. Setup of the user test is illustrated in Figure 6.

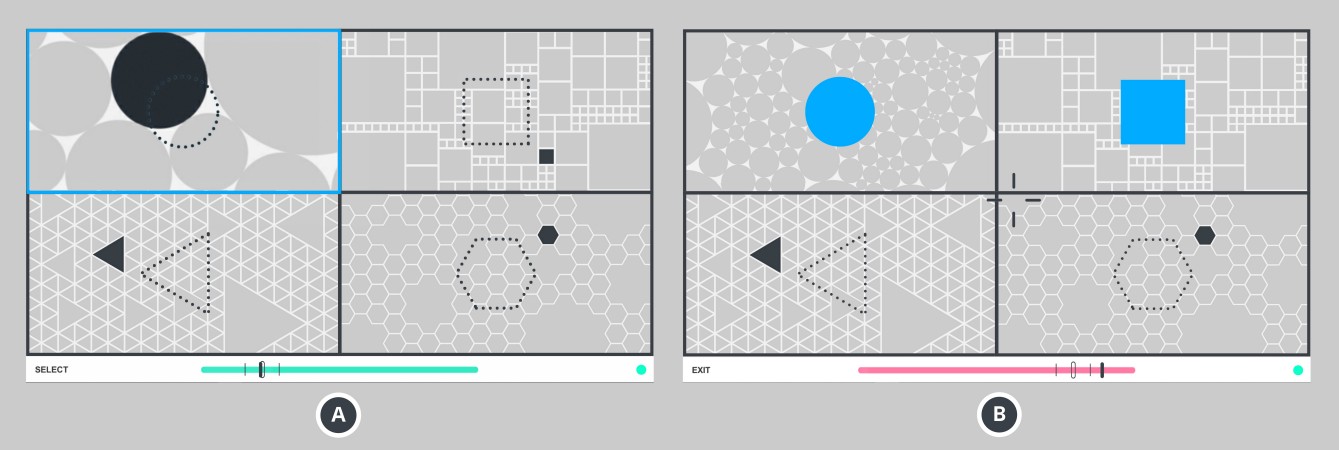

Figure 5: The upper part of the interface is divided into four segments with different geometric shapes. Further down, information on the input mode, a status light (user tracked) and the interactive slider are displayed as visual feedback on the primary task. [A] shows the manipulation mode in which a segment was selected (blue frame) and an image enlargement takes place. In [B] the user is in selection mode and has the possibility to move a cursor to the desired area and select it.

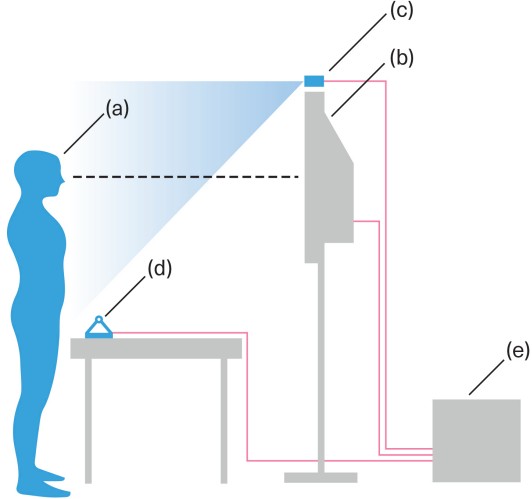

Figure 6: Subject (a) stands in front of a monitor (b) on which a Microsoft Kinect (c) is installed. In between is the control element of the primary task (d) and behind it a computer (e) that processes the data.

### 6.5 Study procedure

After agreeing to the study and collecting demographic data, the presentation of the prototype and the instruction of the individual tasks began. It was explained that the situation of a radiological intervention was simulated by briefly informing the participant about the activities of the radiologist. Then the interface was explained, as well as the various modalities for carrying out the tasks. It was pointed out that the secondary task should be performed as quickly as possible. If the subject was in test group A (with the primary task), the primary task on the physical prototype was explained as an adapted interaction of the catheter navigation task, which should

be performed as accurately as possible. Experimental group B did not receive this explanation.

The system was individually adapted to the person before the actual execution began. Monitor height, height of the physical prototype, handiness (right or left) as well as tracking system was adjusted to the body size. The face also had to be calibrated to allow interaction with the eyebrows.

In order to get a feel for the interaction with the physical prototype and to be able to determine deviations, test group A was used to determine how long they can remain within the tolerance range. The upper part of the interface was hidden, so only the slider was visible. The measurement of the deviation (baseline) took place in a time of 90 seconds. The slider was hidden entirely in test group B to avoid additional distractions.

In the first step before the execution, the control of the cursor was practiced with the help of the head rotation. Information cards for the assigned combinations were placed in the field of view of the participant.

Subsequently, training runs were carried out per combination/interaction technique. If the participants stated to be ready, the first run started and time was measured. After three identical trials, the data (task completion time and time-out) were recorded, and the subject received a NASA TLX questionnaire. The following three combinations and runs were performed according to the same procedure. Finally, a semi-structured interview was conducted. The general feeling during the execution of the experiment was questioned and there was the possibility to give feedback.

### 6.6 Data analysis

All dependent variables were averaged for each of the three trials with identical conditions that each participant encountered. Three-way analyses of variance (ANOVAs) were conducted for the task completion time and the overall TLX rating. A two-way ANOVA was conducted for the error time. The investigator's observations and participants' comments were qualitatively reviewed and clustered by two investigators.

### 6.7 Results

The effects that we found in our inferential statistical analysis are reported in Table 3. We found main effects on the task completion time for the presence of a primary task (Fig. 8) and for the manipulation method (Fig. 7). The manipulation method also showed a

Table 3: Overview of the significant effects found in the evaluation study.

| Dependent variable / effect type | Factor | Degrees of freedom | F-value | p-value | Partial $\eta^2$ |
|---|---|---|---|---|---|
| **Task Completion Time** | | | | | |
| Main effects | **Primary task** | 1 | **8.5** | **0.005*** | **0.132** |
| | **Manipulation method** | 1 | **7.97** | **0.007*** | **0.125** |
| **TLX rating** | | | | | |
| Main effect | **Manipulation method** | 1 | **5.08** | **0.028*** | **0.083** |

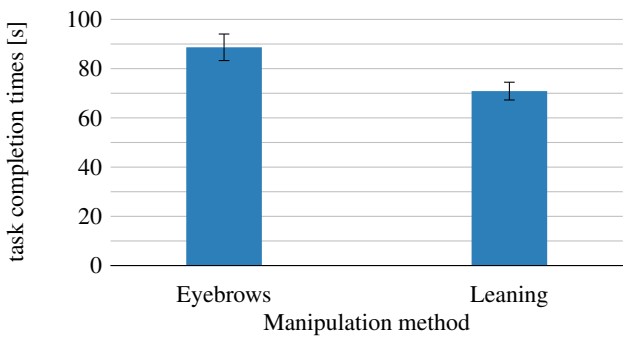

Figure 7: Influence of the manipulation method on task completion time. Error bars show standard error

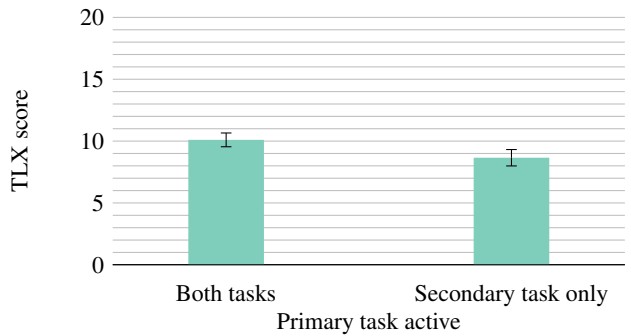

Figure 9: Influence of the presence of a primary task on subjective workload. Error bars show standard error

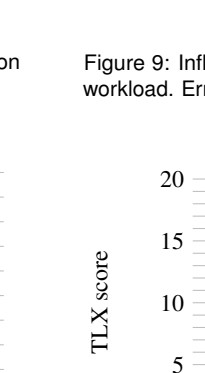

Figure 8: Influence of the presence of a primary task on task completion time. Error bars show standard error

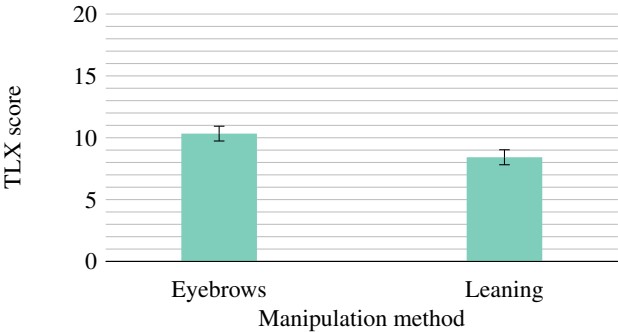

Figure 10: Influence of the manipulation method on subjective workload. Error bars show standard error

main effect on the participants' overall TLX rating (Fig. 10). Within our sample, we observed a trend suggesting that the primary task presence may affect TLX rating (Fig. 9). Although this main effect did not show as significant, the potential effect size is considerable (partial $\eta^2 = 0.049$). The manipulation method showed a significant main effect on the TLX rating. Neither the selection method nor the manipulation method showed significant effects on the error time (i.e. on primary task performance). However, the selection method had a potentially considerable effect size (partial $\eta^2 = 0.084$) that should be investigated further in future studies (Figure 11). We observed no significant interaction effects.

## 7 DISCUSSION

In comparing task completion time between the group with a primary task condition and the one without, we found that performing the secondary interaction takes longer when a primary task has to be fulfilled simultaneously. This is the expected result as additional time is required to switch the focus between both tasks.

As a selection method, head gestures and voice commands were compared for system activation and selection. Pointing was done using head movements in both methods. We found the main effect, within our sample, between the selection method and the time spent outside the target range for the primary task, indicating that head gestures may influence the primary task less than voice commands. The effect was not found to be significant. However, it did show a considerable effect size. Thus, the non-significance may be due to our limited sample size and this possible effect should be investigated further. Comments during the study, however, support an advantage of head gestures over voice commands as nodding and shaking the head are easy to distinguish and to remember compared to voice commands. On the other hand, using speech as an input channel was described as intuitive while shaking the head was found uncomfortable and imprecise.

Taking a closer look at continuous manipulation methods for hands-free zooming reveals that full-body movement (leaning) performed significantly better than facial expressions (moving the eyebrows) in terms of task completion time. The subjective workload was significantly higher for facial expression. A possible explanation

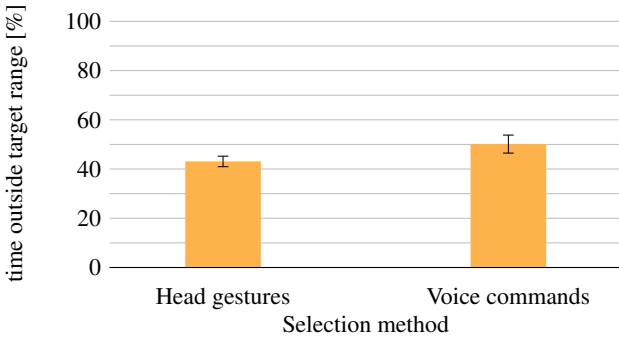

Figure 11: Influence of the selection method on the primary task error rate. Error bars show standard error

can be found in the comments collected during the study. Moving the eyebrows was deemed physically exhausting and according to three participants the fear arose that mood changes would trigger an unintended interaction. Overall, leaning seems to be a suitable, natural input method that allows a faster performance of zooming when the hands are occupied. As a summary, gestures should connect to natural movements as unusual movements such as moving the eyebrows are less precise and exhausting.

Even though a medical scenario served as an exemplary scenario in this work, the tasks were abstracted in a way that no expert knowledge is required. Therefore, the findings presented in this work can be transferred to other domains with similar requirements such as flight coordination, piloting or train driving.

### 7.1 Limitations

The presented study took place in a lab setting, which ruled out external factors and allowed the participants to focus on the given tasks. On the downside, no account was taken for external sources of distraction which might appear in real world scenarios. The abstracted tasks used in the presented study differed from the medical task. The focus for the primary tasks during a catheter intervention lies at the center of the screen instead of the lower edge. Therefore, switching between both tasks might be more demanding in our setup than in the real-world scenario. The primary task baseline was measured at first during the study. Subsequent tasks might have been performed faster due to a learning effect. This effect has to be avoided in the future by applying more extended training periods beforehand and assessing the baseline measures at different points during the study. The position of the Kinect sensor caused minor technical limitations. Eyebrow input was less accurate when the user was leaning towards the display as the eyes and eyebrows were hard to detect at a steep angle. Further, raising and lowering the eyebrows was a discrete interaction compared to continuous leaning. Therefore, eyebrowinput might be more suitable as a safety feature, clutchingmechanism or manipulation method. Speech recognition was accurate in this setup, but in the OR it could be disturbed by conversations among users and sounds from devices present. Especially in the presented scenario, there is a danger that head and body movement can impair the extremely sensitive stability of the catheter. Foot input might be a better choice, but were disregarded as they might interfere with foot pedals as an established modality for controlling medical devices. The interaction between different foot-based input methods needs to be examined to leverage this input channel without affecting the main task.

## 8 CONCLUSION AND FUTURE WORK

In this work, we compared hands-free methods to support concurrent multitasking by using multimodal interaction channels. We found leaning to be a fast method for zooming tasks and gained insights into the suitability of head movements and voice commands as secondary input methods. In the future, the proposed methods need to be compared to domain-specific state-of-the-art input methods such as joysticks, touch screens or task delegation similar to Hettig et al. or Wipfli et al. [8, 26]. Further, the performance at higher cognitive load due to environmental factors, multi-user scenarios or additional types of tasks have to be taken into account. In the long run, the proposed approach might support natural interaction for demanding scenarios, leading to fewer errors and faster task completion.

### ACKNOWLEDGMENTS

This work is funded by the Federal Ministry of Education and Research (BMBF) within the STIMULATE research campus (grant number 13GW0095A).

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
