# OpenReview forum: "Lean-Interaction: passive image manipulation in concurrent multitasking"
_graphicsinterface.org/Graphics_Interface/2020/Conference — GI 2020_

### Official Review · AnonReviewer1 · 2020-01-07
**The paper investigates different types of interactions (head pointing / speech/ facial expressions / full body movement) for point and select and zooming and panning of radiology images on a distant display as a secondary task to a catheter navigation primary task.**

**Confidence:** 4
**Rating:** 3

**Review:**

The paper is not a candidate for acceptance. The first issue is that the paper is not sure whether the work is on solving the specific problem of image navigation and zooming while doing catheter insertion or a generic multitasking scenario. The paper's argumentation is focused on the medical use-case, but then it takes liberties with the inherent assumptions in the medical use-case to argue for a more generic use-case. In the process, the contribution do not make sense either for the specific medical use-case or for the generic multitasking use-case.

Looking at the task from the perspective of the medical use-case, the catheter navigation task is a highly specialized one is the forward-backward simplification is not justified.

Further, the authors disregard the most prominent alternative - using the foot, with some argument about the foot being extremely busy. That is more true for the modalities the paper explores - facial expressions, head pointing, speech, and full body movement are all things that the doctor would need to do in the OR. The explicit start/stop does not make sense for head gestures, since the doctor may nod for other reasons. Further, these gestures run the risk of affecting the steadiness of the catheter - shaking the head can dangerously affect the catheter positioning. A good, physical foot based system seems like a clear and better option any day.

Without the medical usecase, the task specifics do not make sense for generalization to other tasks.

Other points -

1. Why wasn't counterbalancing not done in the evaluation?
2. The charts need to show the performance of all four conditions in one graph.

---

### Official Review · AnonReviewer2 · 2020-01-09
**See review**

**Confidence:** 5
**Rating:** 8

**Review:**

This paper describes a study on multimodal interaction, investigating the effects of a primary task on a secondary (navigation) task. Conditions also look at different modalities for selection zooming, drawn from observations of the target domain based on a medical task.

I enjoyed reading this paper, which is quite well written. The motivation and method are clearly explained. In particular, the introduction does a good job of explaining how the goals are approached by including an exemplary task for observation, which is then abstracted for use in a controlled study. The approach seems like a suitable way to address the research questions about the relationship between the primary and secondary task. The discussion of multi-modal interaction is well-grounded in the literature.

The study design and analysis are well done. My only comment is that the choices of interaction modalities seem somewhat arbitrary. Although I like the idea of basing these from observations, there is no clear connection made to the facial and head gestures, which seem to have been introduced ad-hoc. There is nothing particularly bad about this either, but there are many more potential options that could then be explored as well, so the choices need to be clearly justified. The authors may also be interested to look at work on proxemic interaction, which introduces similar concepts to the leaning. This and other prior work could provide alternate motivations for choosing the modalities.

My only other comment is that is would be good to include additional figures of the task, to better demonstrate the various steps, which are a bit challenging to follow from the text alone.

---

### Official Review · AnonReviewer3 · 2020-01-09
**Well structured and easy to read. However, the necessity of task abstraction and usefulness of the evaluation outcomes are unclear.**

**Confidence:** 4
**Rating:** 5

**Review:**

The paper proposes to design natural interaction for secondary tasks where primary task requires bimanual interaction. The goal is to make sure that the task performance of a complex primary task is not impacted when simultaneously executing the associated secondary task. The paper selected interventional radiology to extract actions and abstract them for designing final interactions. The final designs (body movement, facial expressions, head movement, and voice commands) were evaluated by non-experts. The abstracted action and the interactions within the context of interventional radiology form the original contribution of the paper. Overall, the paper is well structured, easy to read, and follow. However, the reasons for abstracting the task and the usefulness of the evaluation outcomes for the abstracted task are not evident. This affects the quality and significance of the work.

The reason for abstracting the original complex task is not sufficiently justified. One explanation given is that this allows for evaluating the interactions by people outside the domain and possibly applying the findings in other domains. The final interactions were not evaluated by radiologists. Also, the findings were not applied and verified in any other domains. Therefore, it appears that the paper fails to establish both the validity of the interactions for Radiologists and the generalizability of the findings for other domains. The benefits of the abstraction process need to be clearer.

The primary abstracted task was designed with the goal of maintaining a participant's "concentration". However, there are other factors that may impact a radiologist's performance: steadyness of the hand and the head or even the head orientation of the Radiologist in relation to the patient. There is no clear indication why only concentration was selected. Were the factors omitted on purpose? The final design may potentially be unsuitable for a radiologist as shaking and nodding their heads could be unsuitable when handling a catheter. This brings me to my next point. Details on the task performance is missing. Without this data, we do not know how many errors occurred in the primary task and when.

As a minor note, the choice of using head movement for pointing was not supported by any of the observations of the expert.

I suggest three possible ways to strengthen the evaluation of the system.
- get experts' feedback on the final design. This will help confirm whether the interaction works. Feedback may also indicate whether the abstraction process needs to be tweaked.
- apply insights to a different domain to establish generalizability of the findings.
- if the need for abstraction cannot be clearly established, choose a generic concurrent multitasking scenario that truly doesn't require domain knowledge to design & evaluate interaction.

---

### Meta-Review · Area_Chair1 · 2020-01-10

**Recommendation:** Accept
**Confidence:** 4

**Metareview:**

This paper has mixed reviews, with R2 in favour of acceptance, R1 against, and R3 marginal but slightly against.

R3 provides some particularly good feedback about some improvements that can be made to the work. However, both R1 and R3 both focus their reviews on particular details and do not discuss their impact on the overall contribution which R2 sees as lying in the abstract study of complementary modalities, based on observations of a domain task. The primary difficulty outlined by R1 and R3 is that the motivation for the connection the domain task and the abstract task is not clearly explained, leading to confusion about where the paper contribution is intended to focus.  I feel this key issue and other issues raised can be addressed with minor revisions. In addition, R2 and R3 both indicate the paper is well structured and written, so overall lean on the side of acceptance.

---

### Decision · Program_Chairs · 2020-01-11

Accept